# A Novel Angiotensin-I-Converting Enzyme (ACE) Inhibitory Peptide from *Takifugu flavidus*

**DOI:** 10.3390/md19120651

**Published:** 2021-11-23

**Authors:** Yongchang Su, Shicheng Chen, Shuilin Cai, Shuji Liu, Nan Pan, Jie Su, Kun Qiao, Min Xu, Bei Chen, Suping Yang, Zhiyu Liu

**Affiliations:** 1College of Chemical Engineering, Huaqiao University, Xiamen 361021, China; suyongchang@stu.hqu.edu.cn (Y.S.); caishuilin@hqu.edu.cn (S.C.); 2Key Laboratory of Cultivation and High-Value Utilization of Marine Organisms in Fujian Province, Fisheries Research Institute of Fujian, Xiamen 361013, China; cute506636@163.com (S.L.); npan01@qub.ac.uk (N.P.); sjscut@126.com (J.S.); qiaokun@xmu.edu.cn (K.Q.); xumin1315@foxmail.com (M.X.); chenbeifjfri@foxmail.com (B.C.); 3Department of Clinical and Diagnostic Sciences, School of Health Sciences, Oakland University, 433 Meadowbrook Road, Rochester, MI 48309, USA; schen5@oakland.edu

**Keywords:** *Takifugu flavidus*, hydrolysis, ACE-inhibitory activity, purification and identification, molecular docking, antihypertensive activity

## Abstract

Alcalase, neutral protease, and pepsin were used to hydrolyze the skin of *Takifugu flavidus*. The *T. flavidus* hydrolysates (TFHs) with the maximum degree of hydrolysis (DH) and angiotensin-I-converting enzyme (ACE)-inhibitory activity were selected and then ultra-filtered to obtain fractions with components of different molecular weights (MWs) (<1, 1–3, 3–10, 10–50, and >50 kDa). The components with MWs < 1 kDa showed the strongest ACE-inhibitory activity with a half-maximal inhibitory concentration (IC_50_) of 0.58 mg/mL. Purification and identification using semi-preparative liquid chromatography, Sephadex G-15 gel chromatography, RP-HPLC, and LC–MS/MS yielded one new potential ACE-inhibitory peptide, PPLLFAAL (non-competitive suppression mode; IC_50_ of 28 μmmol·L^−1^). Molecular docking and molecular dynamics simulations indicated that the peptides should bind well to ACE and interact with amino acid residues and the zinc ion at the ACE active site. Furthermore, a short-term assay of antihypertensive activity in spontaneously hypertensive rats (SHRs) revealed that PPLLFAAL could significantly decrease the systolic blood pressure (SBP) and diastolic blood pressure (DBP) of SHRs after intravenous administration. These results suggested that PPLLFAAL may have potential applications in functional foods or pharmaceuticals as an antihypertensive agent.

## 1. Introduction

Hypertension, which is a common and serious chronic medical condition, is a major risk factor for developing cardiovascular diseases [1]. Hypertension may cause various acute diseases, such as stroke, coronary heart disease, atherosclerosis, and heart failure. It has become a serious threat to human health, impacting as much as 30% of the global adult population [2]. According to predictive statistics from the World Health Organization (WHO, Geneva, Switzerland), this chronic disease will increase in prevalence by up to 29% of the world’s adult population by 2025. The angiotensin-I-converting enzyme (ACE, EC 3.4.15.1) is a dipeptidyl carboxypeptidase that belongs to the zinc metalloenzyme family and plays a crucial role in blood pressure control via monitoring and attenuating hypertension [3]. ACE is a key enzyme in the renin–angiotensin system (RAS) and kallikrein–kinin system (KKS) since it promotes the conversion of angiotensin I to the potent vasoconstrictor angiotensin II, as well as catalyzes the degradation of the vasodilator bradykinin to raise blood pressure [4,5]. Therefore, the inhibition of ACE activity is a useful treatment for controlling hypertension and thus maintaining blood pressure within a normal range. Synthetic ACE inhibitors (ACEIs), such as captopril, enalapril, lisinopril, alacepril, and ramipril, are widely used in the management of hypertension [6]. Unfortunately, these drugs have a series of side effects, including hypotension, cough, taste disturbances, angioedema, skin rashes, and increased blood potassium levels [7]. Hence, numerous studies were carried out to find alternative safe ACE inhibitors from natural foods. At present, many ACE-inhibitory peptides have been screened from the enzymatic hydrolysates of food protein sources, such as milk proteins, marine organisms, meat, and plant proteins [8,9,10,11,12]. To date, more than 5978 types of ACE-I-inhibitory peptides have been reported in the AHTPDB (antihypertensive peptides database). Some were confirmed to exert definite hypotensive effects in spontaneously hypertensive rats (SHRs), as well as in hypertensive human volunteers [13]. Compared with ACE-inhibitory drugs, ACE-inhibitory peptides have minimal toxicity or side effects relative to synthetic ACE inhibitors [8]. Therefore, ACE-inhibitory peptides have been considered as potent alternatives to synthetic ACE inhibitors in the treatment of hypertension.

Marine organisms, which are rich in unique bioactive proteins, have been widely used in the search for ACE-inhibitory peptides. According to previous studies, many novel ACE-inhibitory peptides with efficient antihypertensive effects were obtained from marine organisms. For instance, Ghassem et al. isolated two novel ACE-inhibitory peptides, namely, LYPPP and YSMYPP, from snakehead fish sarcoplasmic protein hydrolysates, with IC_50_ values of 1.3 and 2.8 μmol·L^−1^, respectively [14]. Liu et al. investigated the antihypertensive activity of a protein hydrolysate that was made from *Rhopilema esculentum* using compound proteinase AQ. Four novel ACE-inhibitory peptides were isolated and their IC_50_ values were 8.40, 23.42, 21.15, and 19.11 µmol·L^−1^ [15]. Hao Wu et al. investigated the antihypertensive effects of ACE-inhibitory peptides from shark meat hydrolysates. Four peptides with high ACE-inhibitory activity were purified; their amino acid sequences were Cys-Phe, Glu-Tyr, Met-Phe, and Phe-Glu. Cys-Phe, Glu-Tyr, and Phe-Glu were confirmed to be novel ACE-inhibitory peptides [16]. Lee et al. identified a potent ACE-inhibitory peptide from tuna frame protein (PTFP), which was composed of 21 amino acids: Gly-Asp-Leu-Gly-Lys-Thr-Thr-Thr-Val-Ser-Asn-Trp-Ser-Pro-Pro-Lys-Try-Lys-Asp-Thr-Pro (MW: 2482 Da, IC_50_: 11.28 μmol·L^−1^). PTFP acts as a non-competitive inhibitor against ACE and has an antihypertensive effect after oral administration in SHRs [17].

*T. flavidus*, otherwise known as tawny puffer, is mainly distributed in the inshore waters of the East China Sea, Yellow Sea, and Bohai Bay [18]. The fish is well known for its distinctive defense mechanism of inflating its body with air or water when threatened. Although its liver and ovary contain tetrodotoxin, *T. flavidus* is considered one of the most delicious dishes in China, Korea, and Japan for its extraordinarily palatable flavor. In addition, *T. flavidus* is rich in collagen, proteins, and carbohydrates, and the high content of crude proteins renders it a promising source of bioactive peptides. The current research on *T. flavidus* has focused primarily on artificial breeding techniques, analysis of the genome, and the nutritional composition [18,19,20]. However, there has been little research on the ACE-inhibitory peptides within *T. flavidus*. 

In this study, *T. flavidus* was hydrolyzed using trypsin, pepsin, and alcalase. The potential ACE-inhibitory peptides of the hydrolysate solution were purified and identified using ultra-filtration, Sephadex G-15 gel chromatography, reversed-phase high-performance liquid chromatography (RP-HPLC), and liquid chromatography with tandem mass spectrometry (LC–MS/MS). Furthermore, the ACE inhibition patterns and molecular docking mechanism of the newly extracted ACE-inhibitory peptides were explored.

## 2. Results and Discussion

### 2.1. Degree of Hydrolysis and ACE-Inhibitory Activity of T. flavidus Hydrolysates (TFHs)

Proteases are necessary to release ACE-inhibitory peptides from inactive forms. Different types of proteases yield polypeptides of different compositions and sizes, which can affect their biological activities [21]. In this study, *T. flavidus* proteins were hydrolyzed in three independent reactions, with each reaction using one of three different proteases, and the ACE-inhibitory activities of different concentrations of the TFHs were assessed (Figure 1A). The half-inhibitory concentrations (IC_50_) of the TFHs that were obtained with alcalase, neutral protease, and pepsin were 0.1, 0.5, and 1.6 mg/mL, respectively. The alcalase-hydrolyzed product exhibited the greatest ACE-inhibitory activity, which even increased with increasing hydrolysis time (Figure 1B). These results implied a correlation between the degree of hydrolysis (DH) and the inhibitory activity. 

Enzymatic hydrolysis is the most efficient pathway for producing bioactive peptides [22]. The biological functions of peptides are strongly affected by the used proteases, which have typical specificities and molecular masses [23]. It was reported that alcalase is an efficient enzyme for hydrolyzing fish proteins and releasing the highest number of potential bioactive (including ACE-inhibitory) peptides from the heavy chain proteins actin, collagen, and myosin [24]. In addition, alcalase tends to cleave peptide bonds that bind to aromatic (Phe, Tyr, and Trp) or uncharged branched (Ile, Val, and Leu) amino acid residues, which can significantly improve the ACE-inhibitory activity [25]. In our study, alcalase seemed to be more effective at digesting *T. flavidus* proteins and resulted in higher ACE-inhibitory activity compared with the other two tested proteases. Thus, alcalase was selected for the generation of ACE-inhibitory peptides from *T. flavidus*. 

### 2.2. ACE-Inhibitory Activity of the TFH and Its Ultrafiltrate Fractions

Hydrolysis using alcalase generally yields a mixture of peptides with various sizes and sequences. Ultrafiltration is commonly used to separate the bioactive peptides with different molecular weights (MWs) from the hydrolysate [26]. We fractionated the TFHs using ultrafiltration through 1, 3, 10, 30, and 50 kDa molecular weight cut-off filtration membranes to obtain fractions with MWs < 1, 1–3, 3–10, 10–50, and >50 kDa. The ACE-inhibitory activity gradually increased as the MW of the components decreased (Figure 2A). The ultrafiltrate fraction with MW < 1 kDa had higher inhibitory activity, and thus lower IC_50_ (0.58 mg/mL), than the fractions with MW > 1 kDa (Figure 2A,B). The results indicated that the low-MW peptides were generally more active than high-MW peptides, which was basically in accordance with the previous research [27,28]. It is believed that short-chain peptides can acquire a spatial conformation that allows them to be positioned within the three-dimensional conformation of the ACE, restricting the access of high-MW peptides [29]. Therefore, the <1 kDa fraction of TFHs was chosen for further separation and purification. 

### 2.3. Purification of T. flavidus Peptides

The fraction containing peptides of molecular weight < 1 kDa was subjected to semi-preparative liquid chromatography on a SinoChrom ODS-BP column as a pre-separation process. Through semi-preparative high-performance liquid chromatography, fractions A1–A8 were successively eluted from the column based on their molecular dimensions (Figure 3A). The eight fractions were collected and lyophilized, and their ACE-inhibitory activities were measured. After dilution to a concentration of 1 mg/mL, they displayed ACE-inhibitory activities ranging from 20 to 90% (Figure 3B). Fraction A7, which showed the highest ACE-inhibitory activity (90%), was then separated via Sephadex G-15 gel filtration chromatography into three major fractions (Figure 3C), of which, A7-c displayed the highest ACE-inhibitory activity (IC_50_ = 0.34 mg/mL) (Figure 3D). Fraction A7-c was further separated using RP-HPLC on an analytical C_18_ column, and three major peaks, named A7-c-1 to A7-c-3, were obtained (Figure 3E). As shown in Figure 3F, fraction A7-c-2 showed the highest ACE-inhibitory activity (IC_50_ = 0.24 mg/mL).

ACE-inhibitory peptides can be purified according to their MW, charge, affinity, and polarity [30,31,32]. Purification methods, such as size exclusion chromatography (Sephadex, Sepharose, Superdex, etc.), ion-exchange chromatography (DEAE-cellulose, DEAE-Sephadex, etc.), and RP-HPLC (C_18_, C_18_, and other columns), are commonly used depending on the peptides’ characteristics [33,34,35,36]. These protein purification procedures lead to appropriate separations. In our study, an ODS-BP column was employed for semi-preparative liquid chromatography as a pre-separation process, resulting in good separation efficiency for TFHs; Sephadex G-15 gel filtration chromatography was used to separate peptide fractions based on their MW. As the final step, RP-HPLC purified the peptides according to their hydrophobic character. The procedure of sequential chromatographic methods enabled us to effectively separate the active peptides of the TFHs. These results are consistent with those of other studies that isolated ACE-inhibitory peptides [37,38]. 

### 2.4. Identification of T. flavidus Peptides and Peptide Synthesis

To identify the potential ACE-inhibitory peptide, the most active sub-fraction, namely, A7-c-2, was analyzed using LC–MS/MS and identified with the PEAKS Studio software. As shown in Figure 4A, the amino acid sequence of the peptide was PPLLFAAL (Pro-Pro-Leu-Leu-Phe-Ala-Ala-Leu, MW = 841.05 Da). The peptide had not been reported previously. It was chemically synthesized so that we could identify its ACE-inhibitory activity. The results showed that PPLLFAAL exhibited high ACE-inhibitory activity, with an IC_50_ value of 28 μmol·L^−1^ (Figure 4B). The inhibition of PPLLFAAL was analyzed according to the Lineweaver–Burk plot method. The peptide was co-incubated with various substrate (hippuryl-l-histidyl-l-leucine (HHL)) concentrations and an ACE solution, and the corresponding double-reciprocal velocity–substrate plot is shown in Figure 4C. When the concentration of PPLLFAAL increased, 1/V_max_ increased, whereas V_max_ decreased and K_m_ did not significantly change, which indicates that its inhibition mode may be a non-competitive one. 

Previous studies revealed that amino acid sequence and hydrophobicity play important roles in the ACE-inhibitory activity of peptides. The presence of an aromatic amino acid (such as Pro, Tyr, or Phe) at the C-terminus and aromatic amino acids (such as Ile and Val) at the N-terminus significantly enhance the ACE inhibition [39,40]. Indeed, PPLLFAAL consisted of hydrophobic amino acids at the N-terminus and aromatic amino acids at the C-terminus, and it also had a high content of hydrophobic amino acids. Hydrophobicity can support peptide-binding to a hydrophobic active center of ACE, thereby increasing the inhibitory activity [41]. In addition, hydrophobic peptides consisting of four to nine amino acids were shown to passively pass through cell membranes via transcytosis or para-cellular diffusion [33]. This suggested that PPLLFAAL might easily be absorbed, which allows it to cross the intestinal wall and enter the blood circulation.

### 2.5. Molecular Simulation of the Interaction between Peptides and ACE

The ACE molecule contains three main active site pockets in the ACE catalytic site, named S1, S2, and S1′ [34]. Hydrogen bond interactions play an irreplaceable role in stabilizing the structure of the enzyme–substrate complex; thus, they are essential for the ACE-catalyzed reaction [42]. Our molecular docking studies indicated that PPLLFAAL was bound to a narrow cavity inside ACE with a relatively extended conformation. The cavity had certain hydrophobicity and hydrophilicity, thanks to which, it could interact well with PPLLFAAL (Figure 5A). According to our simulations, the peptide was bound to the active site pocket of ACE through a network of hydrogen bonds and hydrophobic and van der Waals interactions (Figure 5B). In our model, PPLLFAAL formed 10 hydrogen bonds with residues Glu384, Ala354, Gln281, Lys511, Tyr520, His353, His383, His513, Gln369, and Ala356, and 12 hydrophobic interactions with residues Glu162, Phe457, Val380, Phe527, Glu376, Asp377, Leu161, Trp279, Phe512, Ser355, Val518, and Phe391. Namely, two hydrogen bonds were formed in the S1 active pocket (Ala354 and Glu384), five hydrogen bonds were formed with the S2 pocket (Gln281, His353, Lys511, His513, and Tyr520), and there were hydrophobic interactions with the S1′ pocket. The formation of these interactions greatly stabilized the enzyme–peptide complex. Furthermore, because ACE is a metalloenzyme with a zinc ion in the active site, which is coordinated with His348, Glu372, and His344, the presence of Zn(II) plays an important role in ACE inhibition [43]. The Leu3 of PPLLFAAL was coordinated to the Zn(II) ion, which may be the cause of the deactivation of ACE (Figure 5C). The stability of the ACE–peptide complex was studied using molecular dynamics (MD) simulations. Root-mean-square deviation (RMSD) is an important parameter that is used to indicate the stability of an enzyme–peptide system [44]. It reflects the extent to which protein molecules deviate from their initial structure with the peptide during the dynamic simulation. As shown in Figure 5D, the RMSD of the ACE–PPLLFAAL complex exhibited a large transition, from 0.11 nm to 0.15 nm in 3 ns, then it floated around 0.16 nm. The RMSD values below 0.2 nm during the MD simulations revealed that equilibration of the complex system had been achieved [45]. 

With the advent of molecular simulation techniques, molecular docking and MD simulations could be applied to study the structure–activity relationships between inhibitors and ACE at the molecular level. Tu et al. assessed the molecular docking of casein-derived peptides (NMAINP, NMAINPSK, and NMAINPSKE) against ACE using Discovery Studio 2017 R2 software and found that there were a total of five (Lys118, Asp121, Glu123, Glu403, Arg522), six (Lys118, Asp121, Glu123, Ala356, Glu411, Arg522), and ten (Asn66, Lys118, Asp121, Glu123, Arg124, Trp220, Tyr360, Ser516, Ser517, Arg522) kinds of ACE residues that formed H-bonds with NMAINP, NMAINPSK, NMAINPSKE, respectively [46]. Yu et al. evaluated the interaction mechanism of a peptide (NCW) and ACE using Discovery Studio 2019 software and confirmed that the higher inhibitory potency of NCW might be attributed to the formation of more hydrogen bonds with the ACE active site [47]. These findings are consistent with our results. Our simulation suggested that PPLLFAAL may inhibit ACE via interactions with amino acids at the active site and the zinc ion, thereby blocking the catalytic activity of ACE. PPLLFAAL could bind well to ACE and rapidly form a stable ACE–peptide complex. Hydrogen bonds and hydrophobic interactions between ACE and PPLLFAAL played an important role in maintaining the stability of the ACE complexes.

### 2.6. Antihypertensive Activity of the PPLLFAAL on SHRs

The antihypertensive efficacy of PPLLFAAL in vivo was investigated in terms of changes in systolic blood pressure (SBP) and diastolic blood pressure (DBP) after intravenous administration to SHRs. As shown in Figure 6A, the SBP of the control group after the intravenous administration of saline did not change significantly during a 24 h period. Captopril- and PPLLFAAL-treated SHRs exhibited a significant decrease in SBP. Captopril significantly reduced the SBP (from 190 to 151 mmHg at 4 h, *p* < 0.05), which then increased to 161 mmHg at 24 h. The SBP reduction curve that was obtained for PPLLFAAL was similar to that obtained with captopril. It was notable that the PPLLFAAL could maintain lower SBP levels for a longer period compared with the captopril group after intravenous administration. The results indicated that PPLLFAAL substantially reduced the SBP between 2 and 4 h (*p* < 0.05), with the largest decrease in SBP from 193 to 145 mmHg occurring at 4 h. The SBP then began to recover and maintained a level of 154 mmHg at 24 h. In addition, PPLLFAAL could also affect the DBP (Figure 6B). PPLLFAAL could significantly reduce the DBP of SHRs from 135 to 107 mmHg at 4 h (*p* < 0.05), which was then restored to a level of 113 mmHg at 24 h.

ACE inhibition is often thought to play a key role in controlling blood pressure. Prior to this, ACE-inhibitory peptides are often characterized based on in vitro ACE inhibition. However, the relationship between in vitro ACE inhibition and antihypertensive activity is not apparent due to the complex biological factors, such as degradation by the digestive enzymes, the intestinal barrier, and plasma peptidases [48]. PPLLFAAL is a novel ACE-inhibitory peptide that not only showed potent ACE-inhibitory activity in vitro but also showed effective and prolonged antihypertensive effects in SHRs. It indicated that the PPLLFAAL may maintain the high inhibitory activity in vivo and could effectively avoid the degradation in the blood. This result implied that PPLLFAAL could potentially be applied for the development of novel natural antihypertensive products.

## 3. Materials and Methods

### 3.1. Materials and Chemicals

*T. flavidus* were purchased from Fujian Shenhai Food (Zhangzhou, China) and the skins were peeled off and minced with a meat grinder. Alcalase (EC 3.4.21.62), neutral protease (EC 3.4.22.17), and pepsin (EC 3.4.23.1) were purchased from Solarbio (Beijing, China). ACE (EC 3.4.15.1, from rabbit lung), hippuryl-L-histidyl-L-leucine (HHL), acetonitrile (ACN, HPLC grade), captopril (>99% purity), and trifluoroacetic acid (TFA) were purchased from Sigma Aldrich (St. Louis, MO, USA). Formic acid (FA) was supplied by Merck Chemical Company (Darmstadt, Germany). All other chemical reagents were of analytical grade.

### 3.2. Preparation of TFHs

Alcalase, pepsin, and trypsin were applied to investigate which would be the best enzymatic hydrolysis method, with the degree of hydrolysis and ACE-inhibitory activity as assessment indices. The skins of *T. flavidus* were hydrolyzed using different enzymes under the corresponding optimal temperature and pH conditions (alcalase, pH 8.0, 55 °C; neutral protease, pH 7.0, 55 °C; pepsin, pH 2.0, 37 °C), with the same enzyme/substrate ratio of 2000 U/g and hydrolysis time of 5 h. At the end of the hydrolysis, the hydrolysates were heated at 95 °C for 15 min to inactivate the enzymes. The TFHs were then centrifuged at 8000× *g* for 30 min at 4 °C, and the supernatant was collected, freeze-dried, and stored at −20 °C for further analysis. The degree of hydrolysis (DH) was determined by measuring the nitrogen content that was soluble in 10% (*v*/*v*) trichloroacetic acid [17].

### 3.3. Ultrafiltration of the ACE-Inhibitory Peptide

The lyophilized TFH that was obtained with alcalase was dissolved in distilled water (50 g/L) and filtered sequentially using a continuous flow ultrafilter (STAR Biotechnology, Xiamen, China) with five ultrafiltration membranes with molecular weight (MW) cut-offs of 1, 3, 10, 30, and 50 kDa. Six fractions with MWs <1, 1–3, 3–10, 10–50, and >50 kDa were collected and lyophilized. Peptides with different molecular weights were accurately weighed and dissolved in distilled water to make different concentrations for assessing their ACE-inhibitory activity. The components with higher ACE activity were selected for further purification.

### 3.4. Purification of the ACE-Inhibitory Peptide

The <1 kDa fraction of TFHs was pre-separated using semi-preparative RP-HPLC with a SinoChrom ODS-BP semi-preparation column (50 mm × 400 mm, Elite Technologies, Dalian, China). The sample was filtered through a 0.22 μm filter and elutions were performed with a linear gradient of acetonitrile (10–40%) at a flow rate of 1 mL·min^−1^ and absorbance wavelengths of 220 nm.

The fraction with the highest ACE-inhibitory activity was then further purified using gel filtration chromatography according to the method described by Ma et al., with slight modifications [37]. The sample was loaded onto a Sephadex G-15 (0.5 cm × 100 cm) and eluted using distilled water at a flow rate of 5 mL·min^−1^. The sample was separated and monitored at a wavelength of 220 nm using an AKTA pure 25 (GE Healthcare Bio-Sciences, Uppsala, Sweden).

Following gel filtration chromatography, the most active fraction was subjected to further isolation using RP-HPLC coupled with a C_18_ analysis column (Φ4.6 × 250 mm, Waters, Milford, CT, USA). The column was eluted with a linear gradient of solvent B (acetonitrile containing 0.1% TFA) in solvent A (distilled water), changing from 5 to 25% in 25 min at a flow rate of 1 mL/min, and monitored at 220 nm [49]. The separation was repeated multiple times to obtain sufficient samples for the ACE-inhibitory activity assay. The fractions with the highest ACE-inhibitory activity were subjected to liquid chromatography–mass spectrometry (LC–MS/MS) analysis.

### 3.5. Determination of ACE-Inhibitory Activity

The ACE inhibition was measured using a modified spectrophotometric technique according to the method of Jianpeng Li et al. [50]. ACE and the substrate hippuryl-L-histidyl-L-leucine (HHL, 5 mM) were dissolved in 0.1 M Na_2_[B_4_O_5_(OH)_4_] (pH 8.3) with 0.3 M sodium chloride. Then, 50 µL of the sample solution and 150 µL of the substrate HHL were added to a centrifuge tube, mixed, and incubated at 37 °C for 5 min. The ACE solution (50 mU/mL) was added to start the reaction, and the sample was then incubated at 37 °C for 45 min. The reaction was terminated by adding 250 μL of 1 M HCl. The absorbance of the mixture was measured at 220 nm using an RP-HPLC system (Waters, Milford, MA, USA) on a SunFire C_18_ column (4.6 × 250 mm). The mobile phase comprised distilled water/acetonitrile (75:25 *v*/*v*, with 0.5% TFA) with a flow rate of 1 mL/min. External standard hippuric acid (HA) was used to calculate the concentration of HA. Distilled water was used as a control and the ACE-inhibitory activity (%) was calculated according to Equation (1): (1)$$\mathrm{ACE}-\mathrm{inhibitory}\;\mathrm{activity}\;(\%)=\frac{\mathrm{Ts}-T0}{\mathrm{Ts}}\times 100$$
where T_S_ is the peak area of hippuric acid with distilled water and T_0_ is the peak area with the sample.

The IC_50_ value was defined as the peptide concentration of a sample that inhibited 50% of the ACE activity under the assay conditions.

### 3.6. LC–MS/MS Analysis and Identification of Purified Peptide Sequences

The separated peptides, obtained via the RP-HPLC, with the highest ACE-inhibitory activity were analyzed with a Q Exactive mass spectrometer (Thermo Fisher, Waltham, MA, USA). The parameters that were adopted for the instrument and methods were described by Tu et al. The sample was desalted and loaded onto a chromatographic column (75 μM i.d. × 150 mm, packed with Acclaim PepMap RPLC C_18_, 3 μm, 100 Å). Then, 2% ACN (0.1% formic acid, *v*/*v*) and 80%ACN (with 0.1% formic acid, *v*/*v*) were used as mobile phases A and B, respectively. Elutions were performed with a gradient of 6–95% B at a flow rate of 300 nL⋅min^−1^. The raw mass spectrometry test file was searched using PEAKS Studio software(PEAKS Studio Xpro, Bioinformatics Solutions Inc.) for the corresponding database.

### 3.7. Chemical Synthesis of Peptides

The screened and predicted potential ACEI peptide was chemically synthesized at Sangon Biotech Limited Corporation (Shanghai, China) using a solid phase method. The purity of the peptide was 98%, which was verified by HPLC, and the sequence of the synthesized peptide was determined using HPLC–MS/MS. Synthetic PPLLFAAL was used for the validations of the ACE-inhibitory activity in vitro and in vivo and the determination of the IC_50_ values.

### 3.8. Kinetics of ACE Inhibition

The kinetic inhibition models of the *T. flavidus* peptides were tested using the method of Lin et al. [51]. Briefly, the ACE enzyme activities were measured with different concentrations of the substrate HHL (0.5, 1, 2, and 5 mM). The inhibitory pattern was determined using the Lineweaver–Burk plot of the reciprocal of the production speed of HA (*y*-axis) versus the reciprocal of the substrate concentration (*x*-axis). The V_max_ and K_m_ were calculated, respectively, as shown in the *y*- and *x*-axis intercepts of the primary plot.

### 3.9. Molecular Simulations 

Molecular docking was performed to predict the conformation between ACE and PPLLFAAL using Discovery Studio 2019 software (NeoTrident Technology, Beijing, China). The crystal structure of ACE (PDB ID: 1O8A) was downloaded from the Protein Data Bank (PDB, https://www.rcsb.org/structure/1O8A, 20 October 2021). Water molecules were removed and the zinc ion was retained in the ACE model [52]. The structure of PPLLFAAL was generated with Discovery Studio 2019 and the minimized energy was calculated with the CHARMM force field before molecular docking. The types of interactions between ACE and PPLLFAAL were analyzed. The best molecular docking result was selected based on its docking scores and the binding energy value [46].

The initial coordinates of the ACE–PPLLFAAL complexes for molecular dynamics (MD) simulations were chosen using the criterion of the lowest energy of the complex combined with visual inspection [53]. The MD simulations were performed using Amber software(Amber 20, University of California, San Francisco, CA, US) on the Yinfo Cloud Computing Platform (http://www.yinfotek.com/platform, 20 October 2021) with the parameters set according to Jiang et al. [54]. A subsequent MD simulation of 20 ns length was performed to study the equilibrium properties of the ACE–PPLLFAAL complexes.

### 3.10. Antihypertensive Effect In Vivo

Animals were obtained from Vital River Laboratory Animal Technology (Beijing, China), who also performed the blood pressure measurements in spontaneously hypertensive rats (SHRs) with a tail systolic blood pressure (SBP) of over 180 mmHg. Animals were fed a normal diet, and tap water was freely available. SHRs (11 weeks, 240 ± 20 g body weight) were raised at room temperature (25 °C) under a 12 h light/dark cycle. The antihypertensive effect of PPLLFAAL was evaluated by measuring changes in the SBP and DBP after intravenous injection with a single dose of 5 mg/kg body weight. Captopril (5 mg/kg) was used as a positive control, and the negative control group was injected with an equal volume of saline. The SBP and DBP of the rats were measured using the tail-cuff method before administration and at 2, 4, 6, 12, and 24 h post-administration, using a BP-98A blood pressure monitor (Softron Biotechnology, Beijing, China) after heating the SHRs at 37 °C for 30 min in a warming chamber. All the rats received humane care conforming to the National Institutes of Health Guide for Care and Use of Laboratory Animals. This study was approved by the Ethics Committee of Guangdong medical laboratory animal center (No. 20211001).

### 3.11. Statistical Analysis

Statistical analysis was performed using SPSS 19.0 software (SPSS Institute, Cary, NC, USA). Experimental data are presented as means ± standard deviations (SD) and were statistically evaluated using one-way analysis of variance (ANOVA) with Duncan’s test. Origin 9.0 (OriginLab, Northampton, MA, USA) software was used to process the data.

## 4. Conclusions

In this study, the skins of *T. flavidus* were used to prepare potential ACE-inhibitory peptides. The TFHs that were prepared using different proteases displayed diverse ACE-inhibitory activities in vitro. The highest ACE-inhibitory activities were observed in the TFH that were prepared with alcalase. The components that were obtained using its ultrafiltration showed that the low-MW peptides (<1 kDa) conferred the strongest ACE-inhibitory activity. One novel potential ACE-inhibitory peptide (PPLLFAAL) was isolated from the peptide fraction with MW < 1 kDa using semi-preparative HPLC, a Sephadex G-15, and RP-HPLC and identified using LC–MS/MS. PPLLFAAL showed high ACE-inhibitory activity with an IC_50_ value of 28 μmol·L^−1^ and appeared to be a non-competitive inhibitor. The molecular docking and molecular dynamics simulation results indicated that the peptides could firmly bind to ACE at the active site, as well as to zinc ions. Our results indicated that PPLLFAAL exhibited effective ACE-inhibitory activity in vitro, which makes it a potential candidate for the development of functional foods or anti-hypertensive drugs in the future.

## Figures and Tables

**Figure 1 marinedrugs-19-00651-f001:**
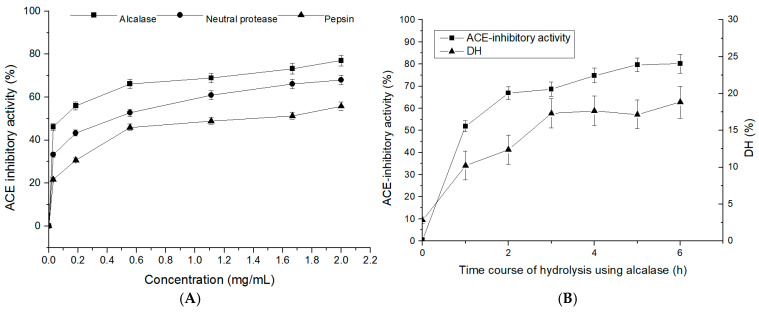
ACE-inhibitory activity of *T. flavidus* hydrolysates (TFHs). (**A**) The ACE-inhibitory activity of TFHs was obtained with alcalase, neutral protease, and pepsin. (**B**) Time course of ACE-inhibitory activity and degree of hydrolysis (DH) of TFHs that were obtained with alcalase.

**Figure 2 marinedrugs-19-00651-f002:**
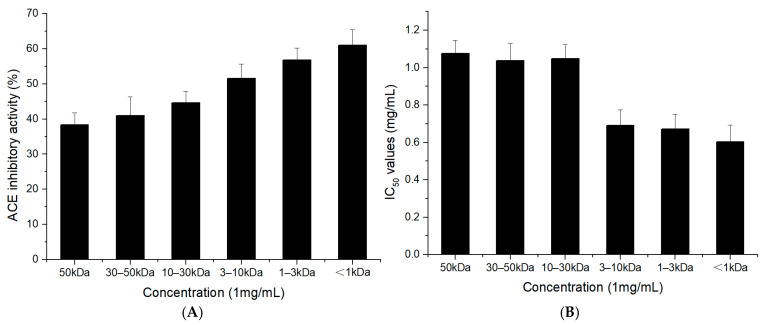
ACE-inhibitory activity of the TFH ultrafiltrate fractions at a concentration of 1 mg/mL (**A**) and corresponding IC_50_ values of the fractions (**B**).

**Figure 3 marinedrugs-19-00651-f003:**
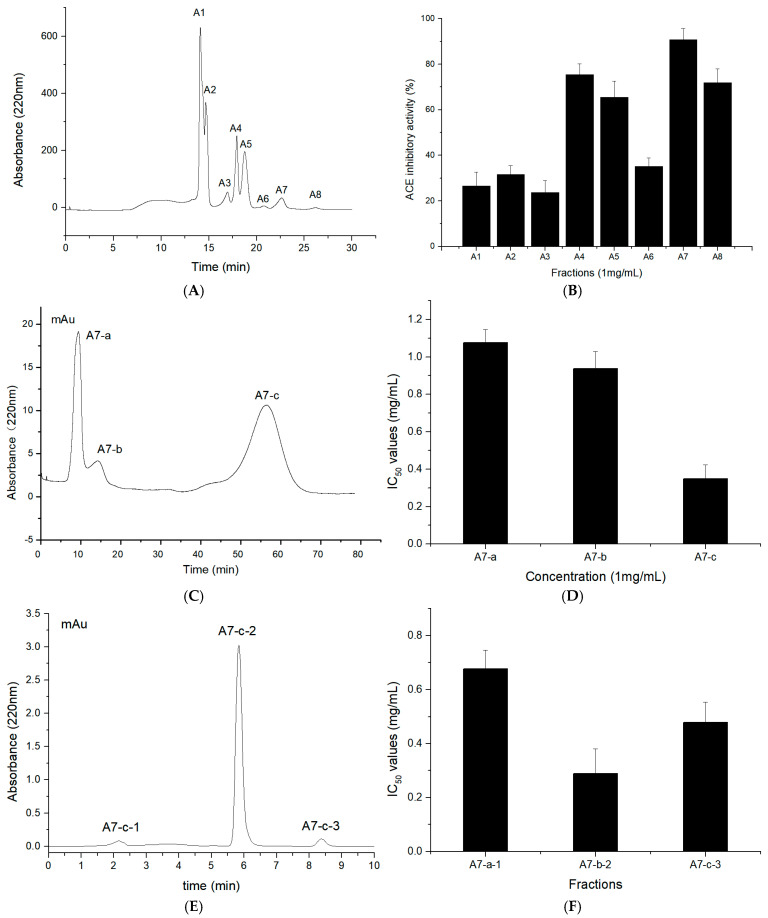
Purification of *T. flavidus* peptides: (**A**) chromatogram of the fractions that were isolated from the fraction of <1 kDa using semi-preparative liquid chromatography, (**B**) ACE-inhibitory activity of the fractions that were obtained via semi-preparative liquid chromatography, (**C**) chromatogram of the fractions that were isolated from A7 using Sephadex G-15 gel chromatography, (**D**) ACE-inhibitory activity of the fractions that were obtained via Sephadex G-15 gel chromatography, (**E**) chromatogram of the fractions that were isolated from A7-c using RP-HPLC, and (**F**) ACE-inhibitory activity of the fractions that were obtained via RP-HPLC.

**Figure 4 marinedrugs-19-00651-f004:**
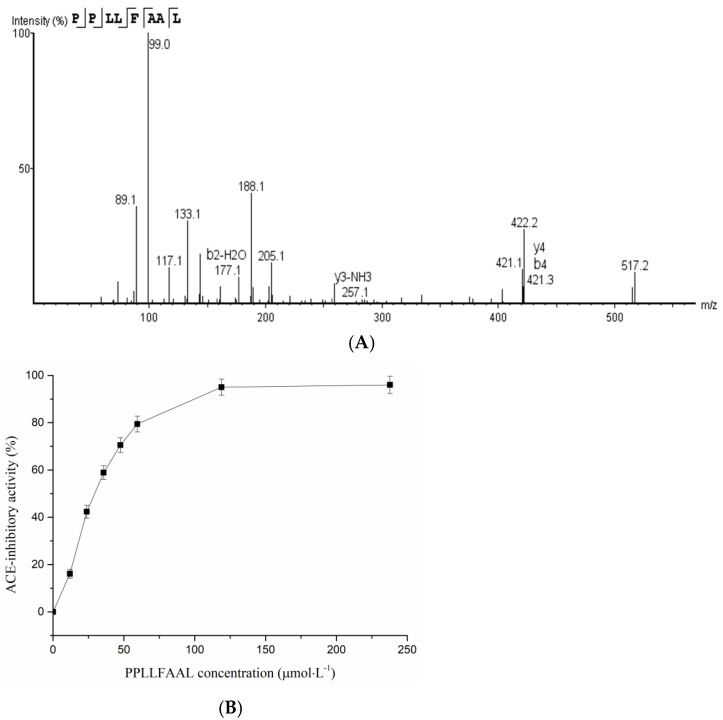
Identification of a *T. flavidus* peptide and its ACE-inhibitory activity: (**A**) MS/MS spectrum of the purified peptide using LC–MS/MS with an ESI source, (**B**) measurement of the ACE-inhibitory activity of PPLLFAAL at different concentrations, and (**C**) the Lineweaver–Burk plots of the reactions of ACE in the presence of PPLLFAAL. [S]: hippuryl-L-histidyl-L-leucine concentration; V: velocity of the reaction.

**Figure 5 marinedrugs-19-00651-f005:**
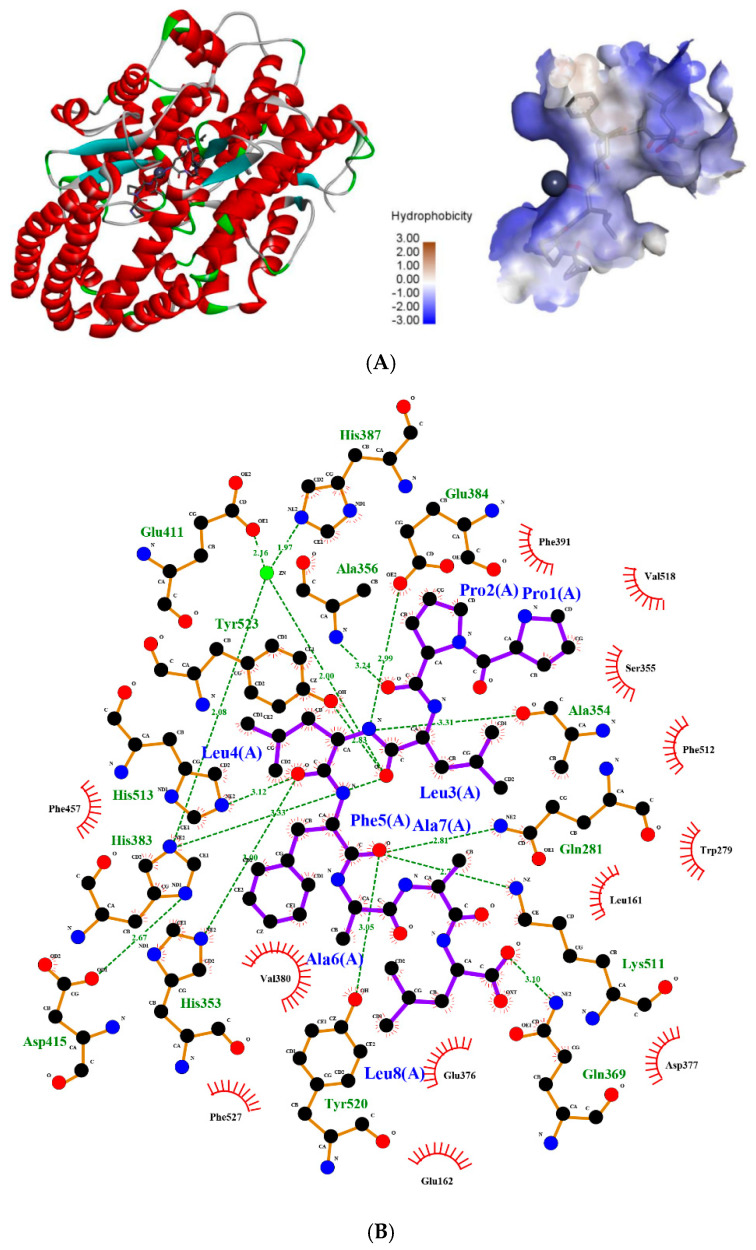
The molecular docking simulations of PPLLFAAL with ACE (PDB: 1O8A): (**A**) general overview and the best-ranked docking pose of peptide PPLLFAAL at the active site, (**B**) the interactions between PPLLFAAL and the residues of ACE, (**C**) the interactions between PPLLFAAL and the Zn^2+^ of ACE, and (**D**) the change in the root-mean-square deviation (RMSD) of the ACE–PPLLFAAL complex over time.

**Figure 6 marinedrugs-19-00651-f006:**
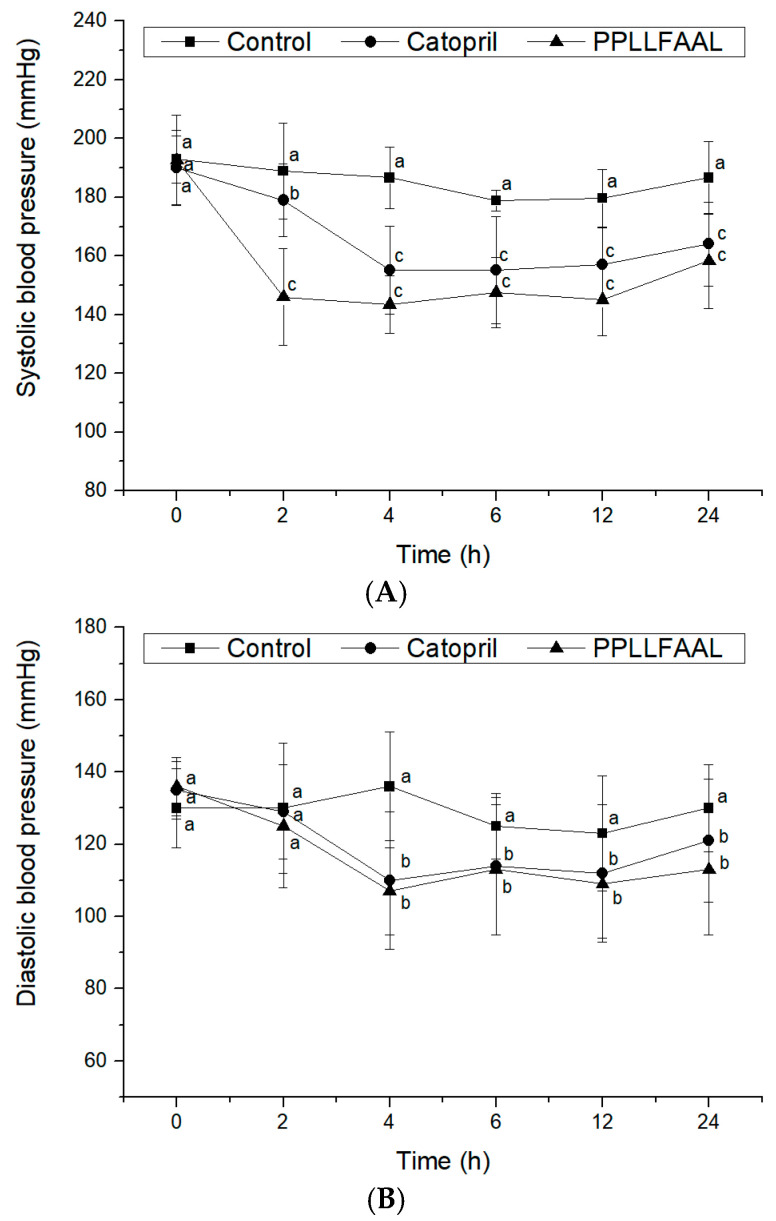
Changes in spontaneously hypertensive rats’ blood pressure after the intravenous administration of PPLLFAAL: (**A**) SBP changes and (**B**) DBP changes. Different letters indicate statistically significant differences, as demonstrated using multiple one-way analysis of variance tests (*p* < 0.05).

## Data Availability

Not applicable.

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
