# Peer review of "A Novel Angiotensin-I-Converting Enzyme (ACE) Inhibitory Peptide from Takifugu flavidus"

_marinedrugs, 2021, doi:10.3390/md19120651_

Round 1

Reviewer 1 Report

Dear Editor.,

Thanks for the invitation to review the manuscript entitled “A Novel Angiotensin-I-Converting Enzyme (ACE) Inhibitory Peptide from Takifugu flavidus”. The concept of this manuscript looks certainly interesting and the manuscript is well-designed and written. I recommend following comments to be accomplished in the manuscript:

Minor revision:

  1. Abstract: The statement “The components of MWs < 1 kDa showed better ACE inhibition” should be modify. What is the mean of better? It should be clarified with IC50.
  2. Preparation of TFHs: which harvesting method has been applied on skin, before hydrolyzing the skins of flavidus?
  3. Also, the origin of PPLLFAAL was nature, but it was better to determine the cytotoxicity of the peptide before in vivo
  4. In this study the antihypertensive effect of PPLLFAAL has been evaluated in Vivo. It is necessary to discuss the in vitro ACE inhibitory and in vivo antihypertensive efficacy of the PPLLFAAL in the animal body.
  5. The in vitro ACE-inhibitory activity of the PPLLFAAL have been confirmed in this study and proposed as a candidate for the development of functional foods. The penetration of the cell membrane and uptake by cells should be discussed according to related references (amino acid characterization and uptake).  
  6. The in vitro and in vivo should be written Italic. The manuscript needs proofreading for some typos or grammatical mistakes.

Reviewer 2 Report

In materials and methods: "Unit" of enzyme should be defined. The same unit is applied to all enzymes?

In eq. 1, A must be defined.

The method of degree of hydrolysis shown in Fig. 1. was not described. 

The amount of peptides used in the assay is required in Fig. 1. In addition, the method used for peptide quantification in Fig. 2 is required.

Results shown in Fig. 3 do not correspond to M&M. No preparative HPLC was described in M&M.

Caption of Fig. 3 is misleading. For example , in fig. E, it was the fraction from gel filtration, not from UF as described.

FIg. 4: Results are not conclusive as only 2 concentrations of the synthetic peptide was carried out.

It was unfortunate that the effect of dose on antihypertension was not studied.

Reviewer 3 Report

Comments:

Manuscript “A Novel Angiotensin-I-Converting Enzyme (ACE) Inhibitory Peptide from Takifugu flavidus” reports some good information on the sample but some parts are not clear in my opinion as I mentioned below. The manuscript is well written. In my opinion before proceeding authors need to justify some points as I brought them as follows:

  1. Page 10, 3.2. Preparation of TFHs: Authors have described the protein hydrolysis. Please clarify that you extracted the protein first or you added the proteases to the whole sample. If you isolated the protein, you need to describe the isolation method carefully.
  1. What method the authors have used to determine the degree of hydrolysis. There is no description or any section to explain how they measure the DH.
  1. There is a DH determination in Figure 1b, but it’s not clear the DH belong to which enzymes and where is the DH analysis for all three enzymes?
  1. Authors have not cleared that the native protein (sample) showed any antihypertensive activity or not. How authors claimed that the bioactivity they determined belong to only hydrolysate. Since they haven’t measured the ACE inhibitory activity for the protein before hydrolysis (parent protein or sample), it is not clear the ACE inhibitory activity belong to protein before hydrolysis or after hydrolysis. If the ACE inhibitory activity belong to the protein before hydrolysis why authors have hydrolyzed the protein. This part needs strong evidence to justify.
  1. The Figure1b shows 5% hydrolysis after 5 h proteolysis time by proteases. What ratio or enzyme concentration authors have used?
  1. In RP-HPLC, how many fractions did you have to evaluate their bioactivity?

Reviewer 4 Report

Comments on the manuscript ID 1444297 by Su et al. entitled “A Novel Angiotensin-I-Converting Enzyme (ACE) Inhibitory Peptide from Takifugu flavidus”:

The authors report а combined experimental/computational study of a potential ACE-inhibitory peptide (PPLLFAAL), isolated from T. flavidus skin, purified and identified. The article is well written, full of content material and beneficial information. The authors logically interpret the data, discuss the results obtained, and draw conclusions.

Minor point:

  • Abstract: the units of IC50 should be corrected (28 μmmol·L-1);

Reviewer 5 Report

The article Su et al., reported ACE inhibitory peptide from Takifugu flavidus skin. They hydrolyzed protein and prepared peptide. In silico analyses were also done.

Many studies of finding ACE inhibitory peptides from natural sources were reported. The most problem of these study is the protein sources of the peptides.

What is the protein sources of this peptide?

What is the main protein in skin?

Author mentioned in introduction that the genome of Takifugu was clear. The authors can find protein sources. Blast search of this peptide cannot find some major protein.

How much peptide can be obtained from the skin? E.g., x mg/1 g skin dry weight?

If the skin contain toxin, can the peptide use for oral drug?

Significance test should be added in Fig. 6.

Round 2

Reviewer 3 Report

The manuscript in current form is acceptable in my opinion

Author Response

The manuscript in current form is acceptable in my opinion

Response: Thank you for your careful review and helpful advice.

Reviewer 5 Report

Authors should reflect your response in your MS.

Response: We thank the reviewer for the careful review of the manuscript.The peptide was analyzed by LC MS/MS and identified with the PEAKS STUDIO software for the T. flavidus protein group database of UniPortKB. The peptide PPLLFAAL was identified from the protein H3CNW0 (Proteinaccession ID in the Uniprot) according to database search results of PEAKS Studio software.

Response: Thank you for the comment. According to our experiment results, the yield of the peptides (MWs <1 kDa) was 360 mg/g of skin dry weight.

I cannot believe your calculation of protein H3CNW0 (360 mg/g dry skin weight) since major protein in skin is collagen.
